# Mitigating CaCO$_3$ crystal nucleation and growth through continuous ion displacement via alternating electric fields

Yiming Liu [1,2,3,11], Minhao Xiao [1,11], Xiaochuan Huang [2,3], Jane Park[4], Matthew E. Hoffman[1], Yuren Feng [2,3], Alicia Kyoungjin An [5], Qilin Li [2,3,6,7], Eric M. V. Hoek [1,8,9,10] ✉ & David Jassby [1,8,9] ✉

Mineral crystal formation poses a challenge on surfaces (e.g., heat exchangers, pipes, membranes, etc.) in contact with super-saturated fluids. Applying alternating currents (AC) to such surfaces can prevent surface crystallization under certain conditions. Here, we demonstrate that ion displacement induced by periodic charging and discharging of the electrical double layer (EDL) inhibits both heterogeneous and homogeneous nucleation (and crystal growth) of CaCO$_3$. Titanium sheets (meant to simulate metallic heat exchanger surfaces) are immersed in super-saturated CaCO$_3$ solutions with a saturation index >11. We show that at relatively high AC frequencies, incomplete EDL formation leads to an alternating electric field that propagates far into the bulk solution, inducing rapid ion migration that overwhelms the Brownian motion of ions. Electrochemical characterization reveals EDL charging/discharging under AC conditions that greatly inhibits precipitation. Operating at 4 V$_{pp}$, 0.1–10 Hz reduces turbidity by over 96% and reduces CaCO$_3$ coverage on the metal plates by over 92%. Based on electrokinetic and crystallization models, the ion displacement velocity (exceeding the mean Brownian velocity) and displacement length disrupts ion collision and crystal nucleation. Overall, the technique has potential for preventing mineral crystal formation in heat exchangers and many other industrially relevant systems.

Mineral crystal formation is a challenge on many surfaces in contact with super-saturated aqueous solutions, such as heat exchangers[1-3], desalination membranes[3-5], and pipes[6,7]. Heat exchangers, in particular, suffer from the formation of a mineral scale on their surface that reduce the heat transfer and overall system energy efficiency[8-10]. Large surface area heat exchangers are exposed to continuous flows of hot and cold streams, which often contain sparingly soluble minerals at super-saturated concentrations[11], some of which have retrograde solubility (e.g., CaCO$_3$), where the solubility of the mineral declines with increasing temperatures (Fig. 1a). Under these conditions, freely

[1]Department of Civil & Environmental Engineering, University of California Los Angeles (UCLA), Los Angeles, CA, USA. [2]National Science Foundation (NSF) Nanosystems Engineering Research Center for Nanotechnology-Enabled Water Treatment, Rice University, Houston, TX, USA. [3]Department of Civil & Environmental Engineering, Rice University, Houston, TX, USA. [4]Department of Chemical & Biomolecular Engineering, UCLA, Los Angeles, CA, USA. [5]Department of Chemical & Biological Engineering, The Hong Kong University of Science and Technology, Hong Kong SAR, China. [6]Department of Chemical & Biomolecular Engineering, Rice University, Houston, TX, USA. [7]Department of Materials Science & NanoEngineering, Rice University, Houston, TX, USA. [8]California NanoSystems Institute, UCLA, Los Angeles, CA, USA. [9]Institute of the Environment & Sustainability, UCLA, Los Angeles, CA, USA. [10]Energy Storage & Distributed Resources Division, Lawrence Berkeley National Laboratory, Berkeley, CA, USA. [11]These authors contributed equally: Yiming Liu, Minhao Xiao. ✉e-mail: emvhoek@ucla.edu; jassby@ucla.edu

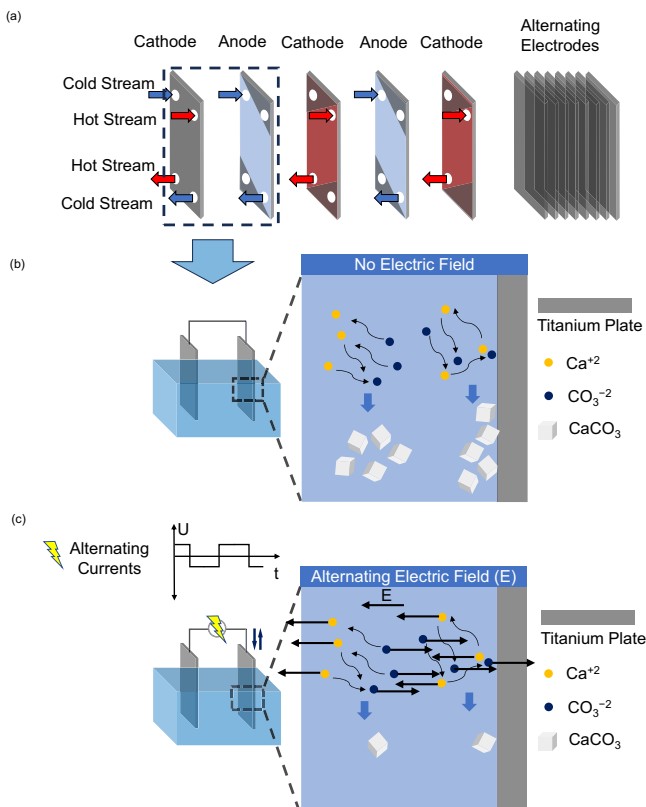

**Fig. 1 | Illustration of heat exchanger system and crystal nucleation prevention mechanism. a** multi-plate heat exchanger arranged to accommodate hot and cold supersaturated streams flowing in parallel. In an idealized configuration, the plates would be arranged as alternating anodes and cathodes. Heterogenous and homogeneous precipitation of CaCO$_3$ without electric field (**b**) and with AC electric field (**c**) in a dual titanium plate system simulating the parallel plates of a heat exchanger.

diffusing ions meet to form nano-sized nuclei, and the nuclei subsequently grow to micro-size crystals. This initial stage of crystallization known as nucleation can take place at the surface/water interface (heterogeneous nucleation) and/or in the bulk solution (homogeneous nucleation) (Fig. 1b). Heterogeneous nucleation is faster than homogeneous nucleation and is particularly relevant to the crystallization processes involving heat exchangers[12]. Nucleation kinetics are described by classical nucleation theory (CNT)[13–15], and supplemented by observations of non-classical nucleation pathways[16–18]. Crystal nuclei or precursors such as amorphous ion clusters form through the collision of ions[19]. In CNT, the rate of nucleation, $J$ (number of crystal nuclei m$^{-3}$ s$^{-1}$), is described by the following Arrhenius-type equation:

$$J = A \cdot \exp \cdot \left( \frac{\Delta G^*}{k_B T} \right) \tag{1}$$

where $A$ (m$^{-3}$ s$^{-1}$) is collision frequency of ions[20–22], $\Delta G^*$ (J) is the free energy barrier that is inversely related to the saturation index[23], $k_B$ (J K$^{-1}$) is Boltzmann constant, and $T$ (K) is the temperature. The free energy barrier of homogeneous nucleation ($\Delta G^*_{homo}$, J) can be explicitly determined by:

$$\Delta G^*_{homo} = \frac{16\pi \gamma_{ln}^3 V_m^2}{3 N_A^2 (k_B T \ln SI)^2} \tag{2}$$

where $\gamma_{ln}$ (J m$^{-2}$) is the interfacial energy between the solvent and the nucleus, $V_m$ (m$^3$ mol$^{-1}$) is molar volume of the crystal nucleus, $N_A$ is the Avogadro constant, and SI is the saturation index. In terms of heterogeneous nucleation, a term for the contact angle between nucleus

and surface ($\theta$, °) is further incorporated into the expression of free energy barrier ($\Delta G^*_{hetero}$, J), which is:

$$\Delta G^*_{hetero} = \Delta G^*_{homo} \cdot \frac{1}{4}(1 - \cos\theta)^2 (2 + \cos\theta) \tag{3}$$

where $\theta$ is greater than 0° but less than 180°[23].

The application of AC to electrically conducting surfaces has been demonstrated to prevent surface crystallization during membrane desalination using electroactive membranes[24–26]. However, the mechanism behind the observed anti-scaling phenomena is not yet fully elucidated, and the observed scaling was assumed to be a result of heterogeneous precipitation only (see Supplementary Note 1 for details). The periodic charging and discharging of the EDL at the surface/water interface occurs in response to the applied electrical potential. There is some evidence that this process induces ion movement both in close proximity to the surface and in the bulk fluid between the electrodes[24,25]. The ion movement helps prevent the formation of mineral crystals[24,25]. This approach towards inhibiting mineral crystal formation is fundamentally different from other approaches, which rely on magnetic fields to manipulate the dipole moment of ions[27–29]. When a direct current (DC) is applied, ions form a permanent EDL on the charged surface such that the concentration of co-ions is depleted and the counter-ion concentration is enhanced[30]. This stoichiometric imbalance between cations and anions within the EDL lowers the local saturation index and increases the free energy barrier for crystal formation[31]. However, due to the thinness of the EDL (a few nm at most) compared to the bulk super-saturated zone, the increase in the free energy, $\Delta G^*$ in Eq. 1, does not significantly disrupt bulk nucleation. In contrast, under AC conditions ion mobility is impacted throughout the space between the polarized electrodes, leading to anti-scaling effects well beyond the boundaries of the EDL[32–34]. The time scale for EDL formation is limited to the AC half-period ($T_{ac}$, s), which is:

$$T_{ac} = \frac{1}{2f} \tag{4}$$

where $f$ (s$^{-1}$) is AC frequency. Therefore, the time to form the crystal nuclei or pre-nucleation ion clusters (i.e., induction time) is constrained by this period, as the ions throughout the space between the electrodes are moving in opposite directions while the EDL is formed. Of course, if the AC period is longer than the time required to form the EDL, ion migration stops, Brownian diffusion takes over ion transport, and nucleation follows the standard model.

At higher polarity reversal frequencies, the EDL transitions to a dynamic state, where it is constantly being formed and disrupted. The incomplete EDL screens fewer surface charges, and leads to the creation of an alternating electric field that propagates far into the bulk and induces fast ion motion in the bulk fluid between the electrodes[35], which can potentially impact both homogeneous and heterogeneous precipitation. It is this dynamic ion motion that can be responsible for the observed dramatic decrease in both surface and bulk crystal formation. The formation and disruption of the EDL is analogous to the charging and discharging of a capacitor via ion migration, based on a Randles equivalent circuit[36]. During the capacitive charging of the EDL, ion migration experiences an initial peak followed by an exponential decay as the EDL is filled up; this can be seen in current/voltage measurements that exhibit a characteristic saw-toothed pattern when AC conditions are applied to electrodes immersed in an electrolyte[37]. Once EDL charging is complete, ion migrations cease, with ion transport only continuing if electrochemical reactions on the electrode consume/generate more charged species. As a result, the time to complete EDL charging, known as EDL charging time, is a critical parameter that determines ion motion. Previous electrokinetic studies

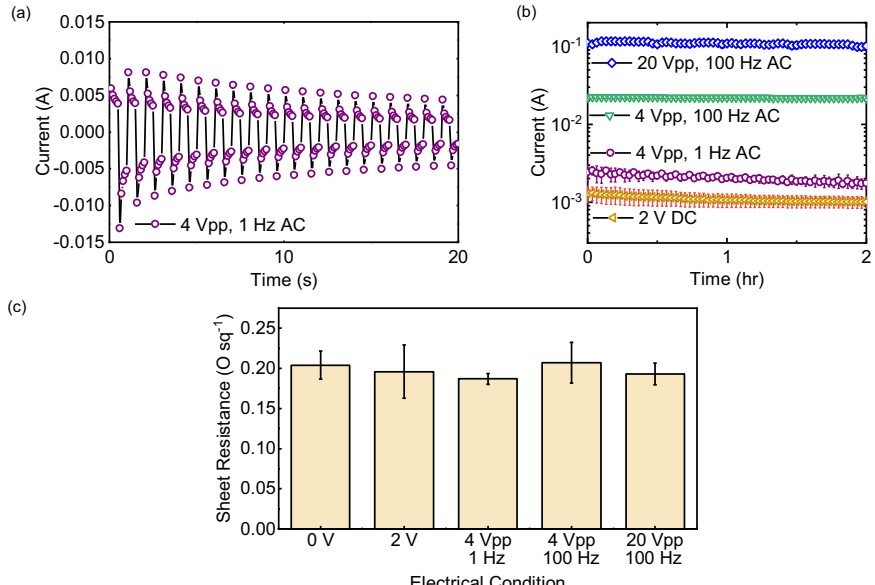

**Fig. 2 | Electrical characterization of electrochemical processes. a** Time-resolved electric current (4 $V_{pp}$, 1 Hz) showing peak current when the polarity is initially reversed, followed by the current decay as the EDL is formed. **b** Averaged current for the different applied voltages and frequencies over time. **c**, Sheet resistance of the electrode surface under different AC, DC, and no potential conditions. Error bars in **b**, **c** represent the standard deviations of triplicated ($N = 3$) measurements.

have shown that the EDL charging time is strongly influenced by ionic composition, the applied voltage, and the external electrical circuit[38,39]. The simplest calculation of the EDL charging time ($\tau_c$, s) under conditions of low applied surface potential (< 0.025 V) is given by:

$$\tau_c = \frac{\lambda_D L}{D} \qquad (5)$$

where $\lambda_D$ (m) is the Debye length that is a function of the solution ionic strength[40], $L$ (m) is the half distance between the electrodes, and $D$ (m² s⁻¹) is the ion diffusivity. Beyond 0.025 V, the accumulation of ions becomes more intense and forms a compact layer next to the electrode surface, which affects the charging dynamics of the EDL. Thus, the EDL charging time is further related to the finite volume of ions and the magnitude of applied potential. In addition, the EDL charging time is also impacted by external resistances present in all electrochemical systems, which create an additional potential drop and impact the actual potential at the surface[41]. To achieve continuous ion migration between two electrodes, an ideal system would apply an AC potential with a frequency such that its half-period is less than the EDL charging time (i.e., $T_{ac} < \tau_c$); in this way, the electric field between the electrodes is never zero, and ion migration in the bulk never decays to zero. The velocity of ion migration ($v$, m s⁻¹) can be calculated as a function of the electric field strength ($E$, V m⁻¹):

$$v = \mu E \qquad (6)$$

where $\mu$ (m² V⁻¹ s⁻¹) is the electrophoretic mobility of specific ion. The distance of ion migration ($L$, m) following the same direction is calculated by:

$$L = \int_0^{T_{ac}} v \, dt \qquad (7)$$

where $t$ is the time after the application of AC potential.

Herein, we report on an AC electrokinetic process that disrupts both homogeneous and heterogeneous nucleation (Fig. 1c). To simulate the crystallization process that would occur between two solid surfaces (e.g., heat exchangers), thin titanium sheets were immersed into super-saturated CaCO₃ solutions. A range of square-wave AC potentials (including DC potentials) were applied to influence crystallization. We characterize the ion motion associated with EDL charging and electrokinetic migration based on electric current and electrode surface resistance. Then, we evaluate the disruption in ion nucleation by tracking the nucleation and precipitation (both homogeneous and heterogeneous) of CaCO₃. We further propose a mechanistic model that calculates ion motion and displacement based on an equivalent circuit and electrokinetic theories, with the model capturing how ion displacement disrupts CaCO₃ nucleation. Last, we validate the model against our experimental data of CaCO₃. Overall, this study demonstrates that an AC potential can reduce the rate of ion collisions to nearly zero. As a result, the collision frequency ($A$ in Eq. 1), is minimized and nucleation is minimized.

## Results

### Characterizing EDL charging and faradaic processes

Here, we explore the impact of the frequency and amplitude of the AC potential on ion movement in the bulk solution. Upon applying an AC potential, the charged surfaces immediately attract counter-ions and repel co-ions. Since ions carry charges, ion flux over the entire cross-section area is equivalent to a flow of charges that balances to the electrical current in the external circuit. Thus, during the formation of the EDL, the electric current serve as an indicator of ion motion[42]. A range of frequencies (0–100 Hz) and potentials (0-20 $V_{pp}$) were examined. Here, 0 Hz is DC, and 0 V is open circuit potential. Also, under DC conditions a potential of 2 V is used, while under 4 $V_{pp}$ AC conditions the potential alternates between +2 V and −2 V (in opposite direction) for each half-period as square-wave. Time-dependent electric current data revealed enhanced ion motion due to EDL charging (Fig. 2a, b). When a DC potential is applied, the current rapidly peaks and quickly decays, illustrating the process of EDL charging (Fig. 2a); the current stabilizes (at very low values) once the EDL charging is complete (Fig. 2b). The current during EDL charging is known as the capacitive current, while the current after EDL charging, typically much lower than capacitive current, is known as the Faradaic current[36]. When an AC potential is applied, the polarity is reversed periodically, forcing the current to flow in the opposite direction (Fig. 2a). Under 4 $V_{pp}$, at 1 Hz, the current decreased from 5.9v6 ± 1.24 to 2.44 ± 0.75 mA

before the polarity was reversed. After that, the current restored to −(6.57 ± 2.09) mA (the negative sign nos the opposite direction) due to the switch of polarity (Fig. 2a). This is a characteristic of the capacitive current responsible for EDL charging, with a half-period that is shorter than the EDL charging time[43]. Note that our observed charging time scale is significantly lower than that observed in capacitive deionization (CDI) systems, which is proportional to electrode capacitance[44]. Typical CDI electrodes have high specific capacitances (i.e., the ability to store charges), a property that is critical to achieve desalination[45]. Under these conditions, the time to fully charge the EDL is longer (i.e., greater EDL charging time scale). To support this statement, we compared the specific capacitance values of CDI systems to the specific capacitance of the electrodes used in our study and found the specific capacitance of CDI electrodes was generally orders-of-magnitude higher compared to our electrode (See Supplementary Note 2 for details).

For a detailed comparison of currents under lower or higher frequencies such as 2 V-DC and 4 $V_{pp}$, 100 Hz, we focused on the absolute values of current, treating both directions equally. These currents (in absolute values) were stable over time, without significant declines in each half-period. The 4 $V_{pp}$, 100 Hz currents (21.95 ± 0.68 mA) exceeded the 2 V DC currents (0.79 ± 0.16 mA), reflecting the dominance of capacitive charging at 100 Hz (Fig. 2b). Also, the average current at 4 $V_{pp}$, 1 Hz (2.05 ± 0.21 mA) was higher than the 2 V-DC current (0.79 ± 0.16 mA). Increasing the potential from 4 to 20 $V_{pp}$ at 100 Hz led to an increase in current from 21.95 ± 0.68 mA to 109.42 ± 1.58 mA (Fig. 2b). The real-time voltage data exhibited consistent EDL charging behavior at different frequencies, in line with the current measurements (See Supplementary Note 3 for details).

In terms of electrochemical reactions, titanium can undergo electrochemical oxidation (corrosion under anodic conditions) when immersed in an electrolyte solution, leading to the formation of a thin layer of the far less conductive titanium oxide that creates additional electrical resistance[46]. The impact of operating conditions on the surface properties of the electrodes were evaluated (Fig. 2c). The sheet resistance values of the electrodes were constant (0.2 Ω square⁻¹) after 2 h under all operating conditions, suggesting that electrochemical corrosion of the electrodes did not take place. While electrochemical corrosion is expected (particularly under DC conditions), the very low concentrations of chloride in these experiments (0.8 mmol L⁻¹) potentially prevented this from happening[47].

## Precipitation kinetics in alternating electric fields

The solutions used in these experiments had a saturation index of 11.04 (calculated using Visual MINTEQ), consisting of 0.4 mmol L⁻¹ CaCl$_2$, 0.4 mmol L⁻¹ Na$_2$CO$_3$, and 10 mmol L⁻¹ NaOH. The 10 mmol L⁻¹ NaOH was used to increase the pH to 12 and ensure that all carbonates in the system were in the form of $CO_3^{-2}$, which facilitates CaCO$_3$ precipitation. The total ionic strength of the solution was constant at 12.4 mmol L⁻¹.

Bulk precipitation of CaCO$_3$ via homogeneous nucleation was tracked by monitoring turbidity, pH, and bulk solution conductivity, while heterogeneous nucleation and surface crystal growth was evaluated through visual and microscopic evaluation of the electrode surfaces. Homogeneous precipitation of CaCO$_3$ was effectively prevented through the application of an AC potential at relatively low frequencies (4 $V_{pp}$ from 0.1 to 10 Hz) compared to all other conditions, over a period of two hours, as measured by the emergence of turbidity (or lack there-of) (Fig. 3a). With no potential applied, the turbidity gradually increased as a result of the nucleation and growth of CaCO$_3$ crystals, reaching a value of 8.75 ± 6.56 NTU after two hours (Fig. 3a). In contrast, the 4 $V_{pp}$ AC potential with a frequency of 0.1, 1, and 10 Hz effectively reduced the final turbidity to 0.26 ± 0.05, 0.21 ± 0.03, and 0.26 ± 0.07 NTU, respectively, over the same two-hour period; more than a 96% reduction compared to the control (Fig. 3a). To verify that

the lack of turbidity in the presence of an electric field was not a result of CaCO$_3$ precipitating elsewhere in the system, a four-hour experiment was performed whereby during the first two hours a 4 $V_{pp}$, 1 Hz potential was applied to the electrodes followed by two-hour period where the potential was turned off. During the initial two-hour period, essentially no crystal formation was observer, whereas during the final two-hour period, the turbidity in the reactor rapidly increased, reaching a value of 5.46 ± 0.92 NTU – very similar to the control (Fig. 3a). This result suggests that the alternating electric potential delayed the formation of CaCO$_3$ crystals, which immediately began forming once the field was removed. However, as the frequency of the applied field (4 $V_{pp}$) increased to 100 Hz, the impact of the applied electrical field disappeared and the turbidity increased to a final value of 6.40 ± 6.03 NTU after two hours, similar to the final turbidity at 0 V (8.75 ± 6.56 NTU). DC conditions were used to investigate the contribution of capacitive adsorption of counterions onto the electrode surfaces to bulk crystal formation. Capacitive adsorption has been used to deionize salt solutions[48] and remove charged aqueous contaminants[49], which can possibly reduce the bulk ion concentration and reduce saturation. However, our experiments show that under DC conditions, the turbidity of the solution showed nearly identical behavior to that of the control (0 V), and had no impact on homogeneous nucleation and bulk crystal formation of CaCO$_3$ (Fig. 3a).

The crystallization was successfully inhibited at the frequencies from 0.1 to 10 Hz but remained unchanged at a higher frequency of 100 Hz. Consequently, we explored another electrical parameter of the AC potential, specifically voltage, to enhance the migration of ions that can disrupt the nucleation and crystallization. Increased voltage results in stronger electric field, which in turn increases the migration velocity of ions. To investigate the contribution of the applied voltage on crystallization, the frequency of the field was fixed at 100 Hz, while the voltage was increased to 12 and 20 $V_{pp}$ (recall that at 4 $V_{pp}$, 100 Hz no reduction in crystallization was observed) (Fig. 3b). At the highest voltage applied (20 $V_{pp}$), the final turbidity (after two hours) was measured at 1.56 ± 1.72 NTU, achieving a modest disruption of CaCO$_3$ crystal formation. However, at lower voltages (12 $V_{pp}$ and 4 $V_{pp}$), turbidity measurements were near those of the control (0 V) (Fig. 3b). This implies that the higher ion migration velocity induced by the 4 and 12 $V_{pp}$ conditions was not sufficient to effectively inhibit CaCO$_3$ crystallization. We suggest that another factor other than migration velocity might also be responsible for the observed scale prevention phenomenon, namely, the migration distance. The ion migration distance between polarity switches is a function of the migration velocity and the time period where the migration takes place (e.g., the half-period of the applied signal). Therefore, at a given applied voltage, the migration distance is larger at 10 Hz (474 nm) than at 100 Hz (56.7 nm). The distance reflects the extent to which cations and anions are moving away from each other during a half-period, which has been previously linked to mineral precipitation in a study using an alternating electric field to prevent membrane scaling[26]. The impacts of frequency and voltages on homogeneous crystallization are summarized in Supplementary Fig. 2a, b.

Under all electrical conditions tested, both solution conductivity and pH linearly decreased over time (Fig. 3c). The decline in pH and conductivity, unlike turbidity, did not mirror the formation of CaCO$_3$ crystals. Rather, we suggest that changes in these parameters were overwhelmingly driven by the dissolution of atmospheric carbon dioxide into the alkaline solution[50]. As CO$_2$ dissolves into water, alkalinity is consumed, which drives the decrease in solution pH. In addition, a single $CO_3^{-2}$ is generated from the reaction between two OH⁻ and one $CO_2$; $CO_3^{-2}$ has a 63.85% lower ion mobility compared to OH⁻[51], which leads to the decrease in ion conductivity. Nevertheless, over the 2-hour duration of the experiment, the decrease in pH and conductivity were almost the same under the no potential and all the electrical conditions, and showed a relatively limited diffusion of CO$_2$

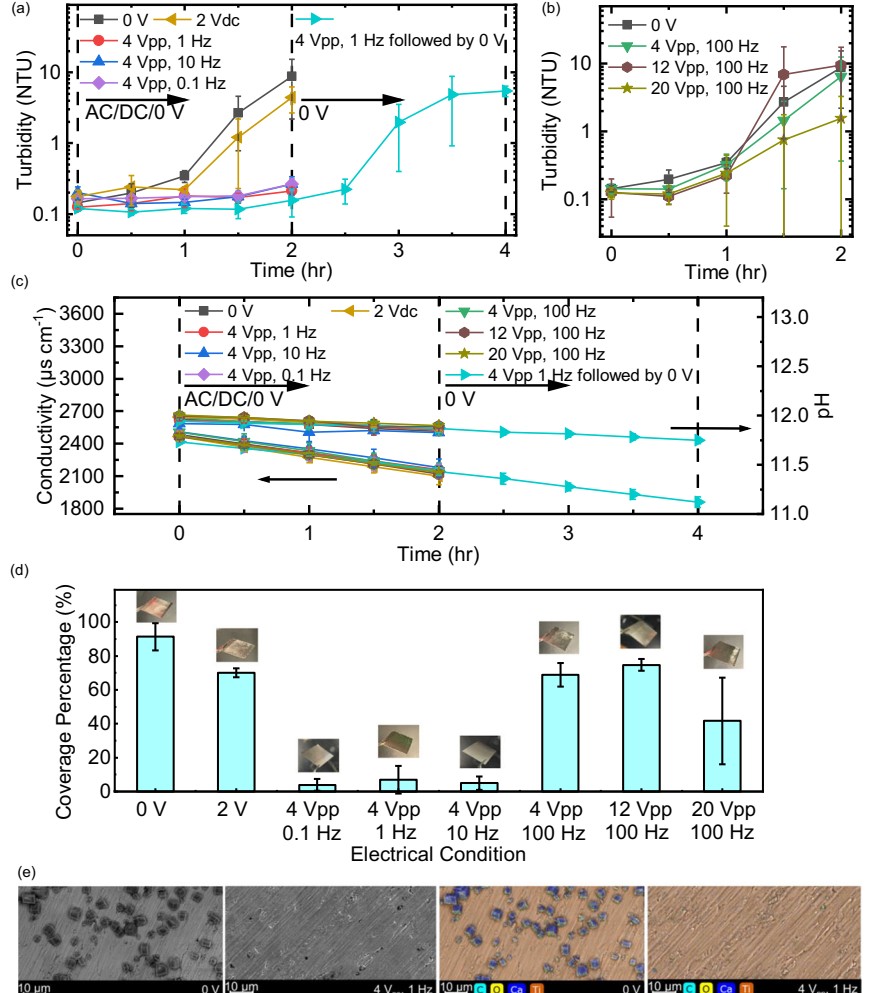

**Fig. 3 | Homogeneous and heterogeneous crystallization of CaCO₃ under the influence of alternating electrical potentials. a** Bulk solution turbidity as a function of time under AC conditions at different voltages and frequencies as a function of time. **b** Bulk solution turbidity under a fixed frequency (100 Hz) and varying applied potentials. **c** Solution conductivity and pH as a function of time under all applied electrical conditions. Lines connecting the datapoints in **a**–**c** are to guide attention, not derived from the model. **d** Electrode % coverage and optical images of electrodes under varying applied electrical conditions. Error bars in **a**-**d** represent the standard deviations of triplicated ($N = 3$) samples of bulk solution and titanium plates. **e** SEM and EDS images of the electrode surface under 0 V and 4 $V_{pp}$, 1 Hz conditions showing the presence and absence of CaCO₃ crystals.

into the solution, which showed negligible impact on CaCO₃ crystallization. We further extended the duration to 24 hours and evaluated the long-term impact of AC potential (i.e., 4 $V_{pp}$, 1 Hz) on turbidity, pH, and conductivity. The mitigation of homogeneous precipitation remained consistently effective and was not affected by CO₂ diffusion into the supersaturated solution (See Supplementary Note 5 for details). Over the 24 h period, the turbidity with AC potential increased only slightly from $0.09 \pm 0.01$ to $0.29 \pm 0.03$ NTU, while the control (0 V) saw a dramatic increase from $0.09 \pm 0.01$ to $80.70 \pm 2.32$ NTU (Supplementary Fig. 3).

To evaluate heterogeneous nucleation of CaCO₃ on the electrode surface, surfaces were extensively characterized for crystal coverage (Fig. 3d,e). Overall, the 4 $V_{pp}$, 1 Hz conditions resulted in the least CaCO₃ electrode coverage, with only $7 \pm 8\%$ of the immersed electrode area covered by crystals after 2 hours of exposure (Fig. 3d). Under other operating conditions, a significantly larger area of the electrode was covered by CaCO₃ crystals: $91 \pm 8\%$ at 0 V, $70 \pm 2\%$ at 2 V DC, $69 \pm 8\%$ at 4 $V_{pp}$, 100 Hz, and $42 \pm 26\%$ at 20 $V_{pp}$, 100 Hz (Fig. 3d). Interestingly, CaCO₃ coverage showed a minimum at 1 Hz, but increased under both lower frequency (DC conditions (0 Hz)) and a higher frequency (100 Hz). By increasing the voltage from 4 to 20 $V_{pp}$

at 100 Hz, CaCO₃ coverage was reduced (from $69 \pm 8\%$ to $42 \pm 26\%$), but was still far higher than under optimal conditions (4 $V_{pp}$, 1 Hz). The findings of crystal coverage align with turbidity measurements that also vary with frequency (0 to 100 Hz) and voltage (4 to 20 $V_{pp}$) (Supplementary Fig. 2a, b). The correlation between surface crystal coverage and turbidity suggests that a unified mechanism related to the EDL's formation and disruption inhibits both heterogeneous and homogeneous CaCO₃ nucleation. Furthermore, this mechanism's influence on ion displacement spans the entire super-saturated area, extending well beyond the EDL's boundaries.

Scanning electron microscope (SEM) images and energy-dispersive X-ray spectroscopy (EDS) mapping were used to compare the structure and elemental composition of electrode surfaces under open-circuit (0 V) and 4 $V_{pp}$, 1 Hz conditions (Fig. 3e). SEM images show that at 0 V, the surface was covered by cubic-like calcite crystals. In contrast, when 4 $V_{pp}$, 1 Hz potentials were applied, very few CaCO₃ crystals were visible, and only titanium was detected. In addition, the % area of CaCO₃ (defined as the ratio of the area of CaCO₃ crystals to the area of the SEM micrograph) was found to be consistent when comparing the SEM micrographs of the same magnification across different electrical conditions (Supplementary Fig. 4a, b). The pure titanium

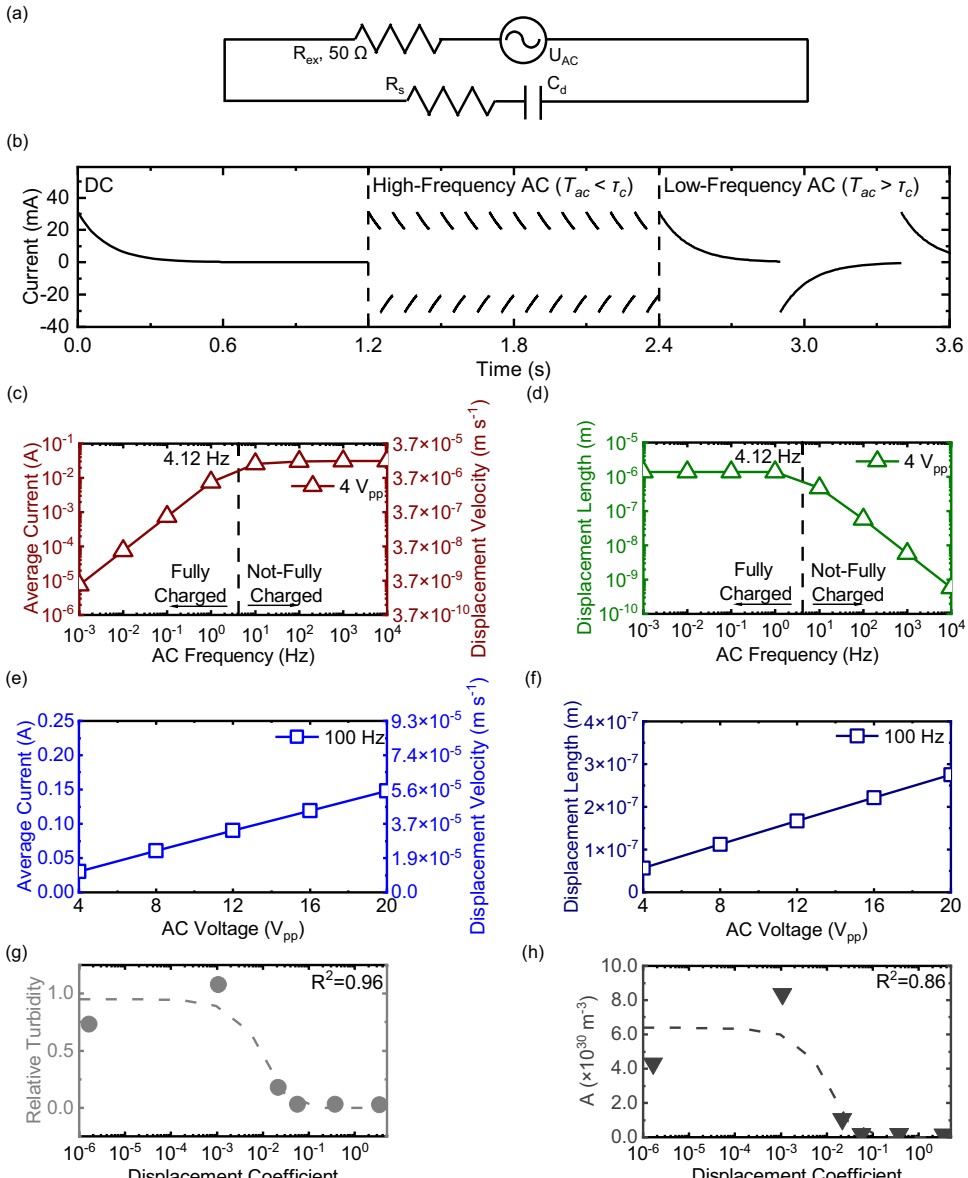

**Fig. 4 | Modeling approach used to relate ion displacement length and velocity to the ion collision frequency through the observed reduction in turbidity under different electrical conditions. a** Equivalent circuit of the experimental system. Combined ideal AC power source ($U_{AC}$) and external resistance ($R_{ex}$) represents AC power supply, $C_d$ denotes EDL capacitance, and $R_s$ denotes solution resistance. **b** Current versus time at DC, and high-frequency and low-frequency AC potentials. Average current and displacement velocity (**c**), and displacement length

(**d**) versus frequency; a reference line is shown for charging frequency. Average current and displacement velocity (**e**), and displacement length (**f**) versus voltage. Lines connecting the datapoints in **c**–**f** are to guide the attention, not derived from the model. Experimentally measured turbidity (**g**) and experimentally derived collision frequency (**h**) versus displacement coefficient. Dashed lines represent fitted function between simulated turbidity/collision frequency and displacement coefficient.

signal in the overall EDS mapping image further confirms the inhibition of CaCO$_3$ crystal formation (Fig. 3e). The individual EDS mapping results and spectrum showed a clear distribution of all the elements on the electrode surface (see Supplementary Note 6 for details).

## Modeling ion displacement and nucleation disruption
We performed model simulations to understand the mechanism of nucleation disruption at the microscopic level. The simulations involved modeling the current, calculating the ion displacement velocity (i.e., ion motion) (Eq. 6) and displacement length (Eq. 7), and correlating them to nucleation inhibition. To model the current, our Randles equivalent circuit was simplified as a composite RC circuit (i.e., resistor in series with capacitor) (Fig. 4a), which excluded relatively insignificant Faradaic processes (compared to capacitive charging

process) (Fig. 2b). The EDL charging time ($\tau_c$, s) was calculated based on EDL capacitance determined by modified Poisson-Boltzmann equations (Eqs. 9–12)[32,38,39], solution resistance derived from ionic properties (Eq. 13), and external circuit resistance (50 Ω, the output impedance of the power supply). As a result, the EDL charging time was computed as 0.121 s at a 4 $V_{pp}$ AC potential (Eq. 14). Following the same calculation, the charging time for a range of voltages are listed in Table 1. In addition, the values of displacement velocity and displacement length at the same frequency where AC half-period equals EDL charging time (i.e., charging frequency introduced below) are also listed in Table 1. By comparing the EDL charging time to the half-period of the applied AC frequency we defined scenarios for DC, high-frequency AC (i.e., $T_{ac} < \tau_c$), and low-frequency AC (i.e., $T_{ac} > \tau_c$) (Fig. 4b). Under DC conditions, the current quickly declines to

**Table 1 | EDL charging time of CaCO₃ super-saturated solution at different applied voltages**

| AC potential ($V_{pp}$) | EDL charging time (s) | Charging Frequency (Hz) | Displacement velocity (m s⁻¹) | Displacement Length (m) |
|---|---|---|---|---|
| 4 | 0.121 | 4.12 | $7.32 \times 10^{-6}$ | $8.88 \times 10^{-7}$ |
| 12 | 0.064 | 7.75 | $2.20 \times 10^{-5}$ | $1.41 \times 10^{-6}$ |
| 20 | 0.049 | 10.16 | $3.67 \times 10^{-5}$ | $1.80 \times 10^{-6}$ |

near-zero, consistent with no capacitive current (only low Faradaic currents). Under AC conditions, the current is consistently restored to the initial capacitive value, which leads to high average currents (Fig. 2b). The average current was calculated for a given EDL charging time, and a given AC voltage and frequency (Eqs. 15 and 16). The predicted currents were in agreement with the measured values (Supplementary Fig. 6).

We further calculated the microscopic displacement of ions based on currents and evaluated the impact of ion displacement on nucleation. The displacement velocity was calculated from the simulated currents, using the one-dimensional Nernst-Planck equation (Eq. 17). After simplifying the Nernst-Planck equation based on our experimental conditions, the current is related to the displacement velocity during electromigration (Eq. 18) (see Supplementary Note 8 for details). The displacement length was determined as a product of the displacement velocity and half-period for each applied voltage and frequency evaluated (Eq. 7). Sufficient displacement of $Ca^{+2}$ and $CO_3^{-2}$ ions impacts nucleation kinetics by preventing the ions from meeting each other and forming crystal nuclei. According to Eq. 1, a reduction in the nucleation rate can be achieved by decreasing saturation index and/or collision frequency. However, the decrease in ion conductivity during experiments was not significantly different between no potential and AC potentials (Fig. 3c), indicating that carbon dioxide diffusion and carbonate concentrations were consistent at all the electrical potentials. Accordingly, the product of the concentrations of $Ca^{+2}$ and $CO_3^{-2}$ divided by the solubility product, equal to the saturation index, was also consistent. Thus, saturation index (and the free energy barrier) is unlikely a factor that leads to crystal formation mitigation. In contrast, the collision frequency, which depends on the relative velocity between oppositely charged ions (i.e., $Ca^{+2}$ and $CO_3^{-2}$) and pre-critical ion clusters, is a likely factor that was impacted by ion displacement.

We propose that the ion displacement velocity and length act synergistically to reduce the collision frequency of nucleation ($A$ in Eq. 1). A successful collision relies on two particles approaching each other with sufficient velocity and proximity. Here the two particles refer to cations and anions, or positively-charged and negatively-charged ion clusters. The increase in displacement velocity suggests that ion migration dominates under the applied electric field, which overshadows Brownian motion. At an induction time of 60 min (established by CaCO₃ precipitation experiments at 0 V), the mean velocity of Brownian motion is only $6.97 \times 10^{-7}$ m s⁻¹. This induction time is well within the induction time conditions identified for CaCO₃, which are dependent on the saturation index and hydrodynamic conditions in the solution[52]. This is substantially lower than displacement velocity of $7.32 \times 10^{-6}$ m s⁻¹ as documented in Table 1 and derived from Eqs. 19 and 20. A further discussion on the interplay between Brownian velocity and time scale is shown in Supplementary Note 9. Under Brownian motion, ion displacement occurs at random directions and velocities[53]. In contrast, oppositely charged ions and particles subject to an alternating electric field move away from each other, along the electric field lines, which could potentially diminish the probability of collision. In comparison, the displacement length disrupts the co-location of cations and anions. As a result, longer

distances are required for the two particles to travel to a common meeting point. Overall, an increase in either displacement parameter (length or velocity) results in a decrease in the nucleation rate, an increase in induction time[24], and eventually prevents nucleation.

A trade-off between displacement velocity and displacement length is evident when varying the AC frequency (Fig. 4c, d). Figure 4c displays both the average current (on the left y-axis) and the average displacement velocity (on the right y-axis) as a function of AC frequency. The charging frequency is defined as the frequency of an AC potential with the same half-period as the EDL charging time, which is 4.12 Hz for 4 $V_{pp}$ AC potentials. As the frequency increases from 0.1 to 10 Hz, the displacement velocity linearly increases from $2.8 \times 10^{-7}$ to $7.3 \times 10^{-6}$ m s⁻¹ from 0.1 to 4.12 Hz, and then slightly increases from $7.3 \times 10^{-6}$ to $9.5 \times 10^{-6}$ m s⁻¹ from 4.12 to 10 Hz. The displacement length is presented as a function of the AC frequency at 4 $V_{pp}$ (Fig. 4d). When the frequency is below 4.12 Hz, the displacement length of ions remains constant at $1.4 \times 10^{-6}$ m. However, when the frequency increases beyond 4.12 Hz, the displacement length decreases, primarily due to a decrease in the half-period. Relatively high displacement velocity and length coexist between 0.1 to 10 Hz compared to 0 Hz (no electric field in the bulk), where displacement velocity is considered as the mean Brownian velocity (relatively low), and 100 Hz, characterized by minimal displacement lengths. This pattern appears to coincide with the observed reduction in CaCO₃ crystal formation both in the bulk and on the surface of the electrodes also from 0.1 to 10 Hz (Fig. 3a).

Increasing the applied voltage leads to an increase in both displacement velocity and length (Fig. 4e, f). The displacement velocity increases from $1.1 \times 10^{-5}$ to $5.5 \times 10^{-5}$ m s⁻¹ when the voltage increases from 4 to 20 $V_{pp}$ (Fig. 4e). As the displacement velocity is proportional to current, the increase in displacement velocity can be analyzed in terms of current. Based on Eq. 16, the increase in voltage from 4 to 20 $V_{pp}$ not only leads to a linear increase in average current, as dictated by Ohm's law, but also induces an additional increase in current arising from a reduced EDL charging time (from 0.121 to 0.049 s). Note that the frequency condition here is 100 Hz, and its half-period of 0.005 s is less than EDL charging time (Table 1). So, the current is predominantly capacitive, which is a function of not only resistance but also capacitance. The EDL charging time is inversely proportional to the voltage (Eq. 14). Nevertheless, the increase in current as a function of voltage is almost linear (Fig. 4e), so the impact of EDL capacitance time on current is minimal (See Supplementary Note 10 for details). Figure 4f illustrates that the displacement length also increases from $5.7 \times 10^{-8}$ to $2.7 \times 10^{-7}$ m when the voltage increases from 4 to 20 $V_{pp}$, proportional to the increase in displacement velocity. Importantly, the increases in both displacement velocity and length from 4 to 20 $V_{pp}$ modestly enhance the disruption of nucleation, as shown in Fig. 3b, further strengthening the connection between increasing the directional ion velocity/displacement length, and the observed decline in crystal formation.

To characterize the disruption to collision frequency caused by displacement, we propose an empirical displacement coefficient ($D_c$), which is a function of displacement velocity and length (Eq. 21); in this equation, $\alpha$ is a weighting coefficient used to relate the relative importance of displacement length over displacement velocity. The observed relative turbidity (defined in Eq. 22) was related to two coefficients ($a$ and $b$), which were used as fitting parameters to relate the relative turbidity to $D_c$ through an exponential expression (Eq. 23). By fitting the experimental data to the model (Supplementary Table 4), $\alpha$ was determined to be 5, while $a$ and $b$ were found to be −68.03 and 0.951, respectively, with $R^2 > 0.95$ (Fig. 4g). An increase in $D_c$ corresponds to a decrease in relative turbidity and crystal formation. $A$ values (i.e., collision frequency of Eq. 1) were calculated from experimental values of turbidity using an adapted two-stage crystallization model (See Supplementary Note 11 for details)[54] and Eq. 24. The

calculated $A$ values as a function of $D_c$ (Fig. 4h) confirms that an increase in $D_c$ also leads to the decrease in $A$ values, consistent with our proposed mechanism, where increased displacement length and velocity reduces the collision frequency, $A$.

## Discussion

We have demonstrated an electrically-driven method to significantly reduce the nucleation and growth of $CaCO_3$ crystals in super-saturated solutions. By applying an AC potential to titanium plates (simulating heat exchanger plates), ion velocity, and displacement length can be manipulated well beyond the EDL region, into the bulk solution. This manipulation can disrupt both homogeneous and heterogeneous nucleation processes. Our findings revealed that applying a 4 $V_{pp}$ AC between 0.1 to 10 Hz resulted in a substantial (> 96%) reduction in turbidity, which indicated a dramatic decrease in homogeneous nucleation. Similarly, the application of this AC potential led to cleaner electrode surface with far less mineral deposition (> 92%), demonstrating a universal anti-crystallization mechanism. A mechanistic model captures the impact of different AC conditions on ionic displacement velocity and length, which offers a pathway to optimize AC frequency and potential for the prevention of $CaCO_3$ precipitation. Through the strategic application of AC potential, we offer a scalable and effective solution for extending the lifespan and reduced maintenance requirements for metallic surfaces in contact with super-saturated aqueous solutions (such as heat exchangers), thereby addressing a longstanding challenge in process engineering.

## Methods

### Lab-scale experiments

The electrochemical cell contains two rectangular titanium sheets (Grade 2, McMaster-Carr, Los Angeles, CA) as electrodes, submerged a $CaCO_3$ super-saturated solution, which is aimed to simulate the initial stages of crystal formation under different electrical conditions. The dimensions of these immersed electrodes were 3 cm by 4 cm, positioned 1 cm apart. The solution was placed in a 200 mL jacketed beaker (Loikaw, Yancheng, China) and was constantly stirred at 250 rpm using a digital magnetic stir plate (Fristadenlab, Reno, NV). The solution temperature was maintained at 25.0 °C using immersion temperature controller (NESLAB RTE7, Fisher Scientific, Waltham, MA), which circulated coolant in the jacket of the beaker. The $CaCO_3$ super-saturated solutions were prepared from stock solutions to include 0.400 mmol $L^{-1}$ $Na_2CO_3$, 10.0 mmol $L^{-1}$ NaOH and 0.400 mmol $L^{-1}$ $CaCl_2$. The 1.00 mol $L^{-1}$ $CaCl_2$ and $Na_2CO_3$ stock solutions were prepared by dissolving 73.5 g $CaCl_2 \cdot 2H_2O$ (99.0–105.0% purity, Fisher Scientific, Waltham, MA) and 53.0 g $Na_2CO_3$ (≥99.5% purity, Fisher Scientific, Waltham, MA) in 500 mL 18 MΩ deionized (DI) water, respectively. Before each experiment, the beaker was rigorously cleaned to remove any mineral residues by using 0.96–1.00 mol $L^{-1}$ $HNO_3$, diluted from concentrated $HNO_3$ (67.0–70.0% w/w in water, TraceMetal™ Grade, Fisher Scientific, Waltham, MA; oxidizer, corrosive to metals, toxic if inhaled) using DI water. Subsequently, the beaker was rinsed by DI water to eliminate residual acid. The $CaCO_3$ super-saturated solution was then prepared by sequentially adding 99.82 mL DI water, 40.0 μL of 1 mol $L^{-1}$ $Na_2CO_3$ solution, 100.0 μL of 10.0 mol $L^{-1}$ NaOH solution (Thermo Fisher Scientific, Waltham, MA), and 40.0 μL of 1.00 mol $L^{-1}$ $CaCl_2$ solution to the clean beaker.

All experiments were triplicated unless otherwise specified. During each 2 h experiment, AC electric potential was applied to the titanium electrodes by using waveform generator (DG1022, Rigol, Suzhou, China). Turbidity, pH, and ion conductivity were monitored at 30 min intervals. Specifically, an 18.0 mL solution sample was drawn every 30 min from the super-saturated solution, and measured by turbidimeter (2100 P, Hach, Loveland, CO), pH electrode (Orion Star A324, Thermo Fisher Scientific, Waltham, MA), and conductivity electrode (Orion Star A222, Thermo Fisher Scientific, Waltham, MA). After each

## Table 2 | Detailed conditions of $CaCO_3$ experiments

| Species | $CaCO_3$ |
|---|---|
| Electrode dimension (cm * cm) | 4*3 |
| Solution volume (mL) | 200 |
| Solution chemistry | 0.4 mmol $L^{-1}$ $Na_2CO_3$ + 10 mmol $L^{-1}$ NaOH + 0.4 mmol $L^{-1}$ $CaCl_2$ |
| Saturation Index | 11.04 |
| Chemical cleaning (> 2 h) | 1 mol $L^{-1}$ $HNO_3$ |

set of measurements, the solution sample was returned to 200 mL solution. In addition, the real-time current and voltage were measured by digital multimeter (5491B, B&K Precision, Yorba Linda, CA) and oscilloscope (DS1054, Rigol, Suzhou, China). Chronoamperometry experiments were conducted using electrochemical workstation (Interface 1010E, Gamry, Warminster, PA) to capture characteristic capacitive and Faradaic currents, contributing to our understanding of the electrochemical dynamics involved. The detailed conditions for $CaCO_3$ experiments are summarized in Table 2.

### Materials surface characterization

After each 2 h long experiment, the electrode surface was gently rinsed by DI water to remove residual homogeneous precipitates. Image of mineral crystal coverage on electrode surface was captured by a smartphone camera (iPhone 12 Pro Max, Apple Inc., CA). The surface area of coverage was directly measured by a length scale. In addition, the electrode was sampled for SEM imaging and EDS analysis (Phenom Pharos, Nanoscience Instruments Inc., AZ). The surface sheet resistance was measured by 4-point conductivity probe (Mitisubish, MCP-T610, Tokyo, Japan).

### Electrochemical and electrokinetic model

The theoretical framework developed for this study captures the EDL charging dynamics and its association with ion displacement and nucleation kinetics. The EDL charging time is a function of external circuit resistance, solution resistance, and EDL capacitance, which is determined by:

$$\tau_c = (R_{ex} + R_s)C_d \tag{8}$$

where $R_{ex}$ (Ω) is the external resistance of 50 Ω, $R_s$ (Ω) is the solution resistance, and $C_d$ (F) is the EDL capacitance. Specifically, the EDL capacitance needs to be considered as a function of solution resistance both depend on the ionic strength of solution ($I$, mol $m^{-3}$), which is calculated by:

$$I = \frac{1}{2}\sum_{i=1}^{n} c_i z_i^2 \tag{9}$$

where $n$ is the total number of ion species, $c_i$ (mol $m^{-3}$) is the molar concentration of species i, $z_i$ is the valence of species $i$. Based on calculated ionic strength, the thickness of EDL, namely Debye length ($\lambda_D$, m), is given by:

$$\lambda_D = \sqrt{\frac{\varepsilon k_B T}{2e^2 N_A I}} \tag{10}$$

where $\varepsilon$ (F $m^{-1}$) is the permittivity, $e$ (C) is the elemental charge, and $N_A$ (mol$^{-1}$) is the Avogadro constant. Since the applied potential is above thermal voltage (0.025 V), we factored into the steric effects of ions and composite Stern and diffuse layers. Modified Poisson-Boltzmann equations were used to calculate the so-called differential

capacitance[32,38,39]. First, the volume fraction ($\nu$) is determined by:

$$\nu = 2a^3 c_0 N_A \qquad (11)$$

where $a$ (m) is the diameter of ion and $c_0$ (mol m$^{-3}$) is the total molar concentration of ions in the bulk solution. Then, the differential capacitance ($C_d$, F) is calculated by:

$$C_d = \frac{\frac{\varepsilon}{\lambda_D}\left|\sinh\left(\frac{ze\zeta}{2k_BT}\right)\right|\sinh\left(\frac{ze\zeta}{2k_BT}\right)}{\left[1+2\nu\sinh^2\left(\frac{ze\zeta}{2k_BT}\right)\right]\sqrt{\frac{2}{\nu}\ln\left[1+2\nu\sinh^2\left(\frac{ze\zeta}{2k_BT}\right)\right]}} A_e \qquad (12)$$

where $\zeta$ is the applied surface potential (V), $z$ is the average ion valence, and $A_e$ (m$^2$) is the electrode surface area. Solution resistance ($R_s$, $\Omega$) is calculated as:

$$R_s = \frac{\lambda_D^2 L}{\varepsilon D} A_e^{-1} \qquad (13)$$

where $L$ (m) is the distance between electrode to center of bulk solution in a symmetric two-electrode setup, and $D$ is the average diffusion coefficient. The total resistance consists of solution resistance and external resistance ($R_{ex}$, 50 $\Omega$). By substituting Eqs. 12 and 13 into 8, the EDL charging time in the final form is presented as:

$$\tau_c = \frac{\left(R_{ex}\frac{\varepsilon}{\lambda_D}A_e + \frac{\lambda_D L}{D}\right)\left|\sinh\left(\frac{ze\zeta}{2k_BT}\right)\right|\sinh\left(\frac{ze\zeta}{2k_BT}\right)}{\left[1+2\nu\sinh^2\left(\frac{ze\zeta}{2k_BT}\right)\right]\sqrt{\frac{2}{\nu}\ln\left[1+2\nu\sinh^2\left(\frac{ze\zeta}{2k_BT}\right)\right]}} \qquad (14)$$

Both current and voltage are calculated in the equivalent circuit model based on the Ohm's and Kirchhoff's circuit laws. The instantaneous current ($I$, C s$^{-1}$) is determined by:

$$I = \frac{U_{AC}}{R_{ex}+R_s}e^{-\frac{t}{\tau_c}} \qquad (15)$$

where $U_{AC}$ (V) is the applied voltage. By integrating the instantaneous current over an AC half-period ($T_{AC}$, s), the average AC current ($\bar{I}$, C s$^{-1}$) can be calculated as:

$$\bar{I} = \frac{U_{AC}}{R_{ex}+R_s} \times \frac{\tau_c}{T_{AC}} \times (1 - e^{-\frac{T_{AC}}{\tau_c}}) \qquad (16)$$

The conversion from current to ion transport values relies on one dimensional Nernst-Planck equation, which describes the molar flux of ions ($J$, mol m$^{-2}$ s$^{-1}$) as a sum of diffusion ($-D\frac{\partial c}{\partial x}$), convection ($cv$), and migration ($\frac{zFD}{RT}cE$). The exact expression is:

$$J = -D\frac{\partial c}{\partial x} + cv_f + \frac{zFD}{RT}cE \qquad (17)$$

where $D$ (m$^2$ s$^{-1}$) is the ion diffusivity, $c$ (mol m$^{-3}$) is the ion concentration, $x$ (m) is the distance between ion species and electrode, $v_f$ (m s$^{-1}$) is the flow velocity, z is the ion valence, $F$ (C mol$^{-1}$) is the Faraday constant, and $R$ (J mol$^{-1}$ K$^{-1}$) is the ideal gas constant. The average displacement velocity of ions ($v$, m s$^{-1}$) is proportional to the average current ($I$, C s$^{-1}$) (detailed derivation shown in Supplementary Note 8):

$$v = \frac{\frac{I}{A_e F}}{\sum zc} \qquad (18)$$

where $F$ (C mol$^{-1}$) is Faraday's constant, $z$ is the ion valence, and $c$ (mol m$^{-3}$) is the molar concentration of ions. To capture the velocity of Brownian motion, we first used the classical relation by Einstein[53] to calculate the mean squared displacement length ($\Delta x$, m):

$$\overline{(\Delta x)^2} = 2\bar{D}t \qquad (19)$$

where $\bar{D}$ (m$^2$ s$^{-1}$) is the average diffusivity of nucleation cation (i.e., Ca$^{+2}$) and anion (i.e., CO$_3^{-2}$). In this case, the mean displacement velocity of Brownian motion ($\bar{v}_B$, m s$^{-1}$) is given by:

$$\bar{v}_B = \frac{\left(\overline{(\Delta x)^2}\right)^{\frac{1}{2}}}{t} = \left(\frac{2\bar{D}}{t}\right)^{\frac{1}{2}} \qquad (20)$$

Here, the displacement length is a function of the inverse of the time, which is different from the displacement velocity of AC potential independent of the time.

## Model validation

The quantitative relationship between ion displacement and nucleation is established by fitting simulation results to experimental data. Reference values of displacement velocity, displacement length, and turbidity are used for unit normalization and do not affect the goodness of fit (R$^2$). The displacement coefficient is introduced as a bulk parameter that characterizes the disruption to crystal formation, which is determined by:

$$D_c = \left(\frac{v}{v_{ref}}\right)\left(\frac{L}{L_{ref}}\right)^\alpha \qquad (21)$$

where $v_{ref}$ (m s$^{-1}$) is the reference displacement velocity and $L_{ref}$ (m) is the reference displacement length at charging frequency of 4 V$_{pp}$ (Table 1). Relative turbidity that quantifies the prevention of crystal formation under all the electrical conditions were used as an experimental metric, which is expressed as:

$$\text{Relative Tubirdity} = \frac{\text{Final Turbdity (applied potential)}}{\text{Final Turbdity (no potential)}} \qquad (22)$$

where the final turbidity of an applied potential was compared to final turbidity without any applied potential (as a reference value) at the end of the two-hour long experiment. Based on curve fitting, an exponential relationship between relative turbidity and displacement coefficient is established, which is:

$$\text{Relative Turbidity} = ae^{-bD_c} \qquad (23)$$

where $a$ and $b$ are coefficients for the exponential fitting. To have a fundamental understanding between ion displacement and nucleation, the pre-exponential factor of Eq. 1 ($A$), known as collision frequency, is studied. While multiple forms proposed in previous reports[52,55,56], the exact $A$ value is generally determined by fitting the experimental values to the nucleation kinetic equation, which is:

$$\ln J = \ln A - B\frac{1}{(\log SI)^2} \qquad (24)$$

where $B$ is the shape factor, and SI is the supersaturation index.

## Data availability

The experimental and modeling data generated in this study are provided in the main text and Supplementary Information. Source Data file for the figures has been deposited in Figshare under accession code DOI link. [https://doi.org/10.6084/m9.figshare.27855174].

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

## Acknowledgements

This work is supported by the Department of Energy (Award No. DE-EE0008391, D.J.), the California Department of Water Resources (D.J.), the Sustainable LA Grand Challenge of UCLA (E.M.V.H.), and the NSF Nano-Enabled Water Treatment Center (Q.L.). The authors thank all the colleagues in the Jassby and Hoek groups at UCLA and the Li group at Rice University for their support. The authors also appreciate the helpful suggestions from Dr. Xin Chen regarding EDL model and those from Mr. Adriano Leão regarding solution sampling within the Institute of Carbon Management and Department of Civil and Environmental Engineering at UCLA.

## Author contributions

Y.L., E.M.V.H., and D.J. conceived the conceptual idea. Y.L., M.X., X.H., J.P., M.E.H., Y.F., A.K.A., and Q.L. designed and performed the experiments and analyzed the data. M.X. performed the microscopic characterization. Y.L. developed the mechanistic model. Y.L., M.X., E.H. and D.J. drafted and revised the manuscript. All authors discussed the results and commented on the manuscript. All aspects of the research were overseen by D.J.

## Competing interests

The authors declare no competing interests.
