## [Transparent Peer Review file · Nature Communications]

Mitigating CaCO₃ crystal nucleation and growth through continuous ion displacement via alternating electric fields

Corresponding Author: Professor David Jassby

Version 0:

Reviewer comments:

Reviewer #1

(Remarks to the Author)

The authors investigated the ion displacement phenomena in the AC field to reveal the anti-scaling mechanism. The use of AC to mitigate scaling has been reported in a number of publications. The proposed mechanism that AC disrupts ion movement is not new; it was proposed in a previous publication. The main focus of this work is to apply a model to investigate the process in more details. The study is systematic with detailed discussions. However, the basis for discussions might not be well validated. It may be more suitable for a more specific journal. Some comments are as follows:

1. The authors used EDL charging to explain the observations during AC. They correlated AC current with ion concentrations, and used it as an indication of EDL charging. However, in literature work (ref. 42,43), voltage is altered between positive and negative in the time scale of hours, which allows sufficient time to observe the EDL charging (it can be seen from the references that the concentration changes are not achieved in seconds). However, the half time of AC here is in the order of 0.1-0.001 seconds. This is far too short to observe EDL charging. The changing current with time seen in the experiment is due to the increasing/decreasing voltage of AC.
2. Lines 113-115, the statement 'The incomplete EDL screens fewer surface charges, and leads to the creation of an alternating electric field that propagates far into the bulk and induces fast ion motion in the bulk fluid between the electrodes' is confusing. AC leads to frequent change of electric field between the two electrodes, and thus includes fast ion motion. This effect could be direct. It does not have to be linked to EDL and surface charge (the authors claimed so since they used a hypothesis of a capacitor).
3. Conclusion, it was stated that 'we introduce an electrically-driven method to significantly reduce the 436 nucleation and growth of CaCO₃ crystals in super-saturated solutions', and 'Through the strategic application of AC potential, we offer a scalable and effective solution for extending the lifespan and reduced maintenance requirements for metallic surfaces in contact with super-saturated aqueous solutions (such as heat exchangers), thereby addressing a longstanding challenge in process engineering'. This method has already been introduced in a number of other works. Hence, such conclusions are misleading as the authors overstate their contributions.

Reviewer #2

(Remarks to the Author)

The manuscript presents an effective method to prevent CaCO₃ crystal nucleation and growth in supersaturated solutions using alternating electric fields (AC). The results and theory are convincing, demonstrating that AC disrupts ion collisions through rapid ion displacement induced by the charging and discharging of the electrical double layer (EDL) at certain frequencies. The study combines robust electrochemical characterization and modeling to demonstrate that high-frequency AC fields prevent both homogeneous and heterogeneous nucleation without significant electrode corrosion, suggesting practical applications in industrial systems like heat exchangers. This is a significant advancement in the field of mineral scale prevention. Below are some comments for the authors to consider.

Please discuss the novelty of this study further. The method of applying AC potentials to prevent mineral scaling, along with the underlying theory, has been presented in previous papers, including the authors' publication [1, 2].

The turbidity, pH, and conductivity were monitored to indicate bulk precipitation of CaCO₃ within 2 hours. It may be helpful to elucidate the contribution of the dissolution of atmospheric CO₂ into the alkaline solution to the observed changes in pH and

conductivity over 2 hours. Could the authors extend the duration of the experiment, which might allow for a clearer observation of the effects of AC potentials on CaCO₃ crystallization, as well as any potential long-term impacts of CO₂ diffusion? This could help differentiate between short-term and long-term effects and provide more comprehensive insights into the mechanisms.

On page 14, Figure 3d, the authors estimated the percentage of CaCO₃ electrode coverage and claimed that increasing the voltage from 4 to 20 V_{pp} at 100 Hz reduced CaCO₃ coverage (from 69±8% to 42±26%). However, the scales under 20 V_{pp} look denser than those under other experimental conditions. More quantification of the scales should be considered.

Reference

[1] U. Rao, A. Iddya, B. Jung, C.M. Khor, Z. Hendren, C. Turchi, T. Cath, E.M. Hoek, G.Z. Ramon, D. Jassby, Mineral scale prevention on electrically conducting membrane distillation membranes using induced electrophoretic mixing, *Environ Sci Technol* 54(6) (2020) 3678-3690.

[2] B. Jung, S. Ma, C.M. Khor, N.K. Khanzada, A. Anvari, X. Wang, S. Im, J. Wu, U. Rao, A.K. An, Impact of polarity reversal on inorganic scaling on carbon nanotube-based electrically-conducting nanofiltration membranes, *Chem Eng J* 452 (2023) 139216.

Version 1:

Reviewer comments:

Reviewer #1

(Remarks to the Author)

The authors have addressed questions raised by reviewers.

Reviewer #2

(Remarks to the Author)

The authors have addressed my comments and provided additional data. I would like to recommend the acceptance of the manuscript.

Reviewer #1:

Reviewer: *The authors investigated the ion displacement phenomena in the AC field to reveal the anti-scaling mechanism. The use of AC to mitigate scaling has been reported in a number of publications. The proposed mechanism that AC disrupts ion movement is not new; it was proposed in a previous publication. The main focus of this work is to apply a model to investigate the process in more details. The study is systematic with detailed discussions. However, the basis for discussions might not be well validated. It may be more suitable for a more specific journal. Some comments are as follows:*

Authors: We thank the reviewer for pointing out the use of AC fields in mitigating mineral scaling. While similar mechanisms have been explored, our study introduces a novel approach by successfully preventing homogeneous precipitation, a finding not reported in previous AC anti-scaling research. This new insight into preventing both homogeneous and heterogeneous nucleation of CaCO₃ crystals is a significant contribution. Furthermore, we move beyond empirical observations to establish a mechanistic rationale for the observed phenomena, which has not been done before.

Reviewer: *1. The authors used EDL charging to explain the observations during AC. They correlated AC current with ion concentrations, and used it as an indication of EDL charging. However, in literature work (ref. 42,43), voltage is altered between positive and negative in the time scale of hours, which allows sufficient time to observe the EDL charging (it can be seen from the references that the concentration changes are not achieved in seconds). However, the half time of AC here is in the order of 0.1-0.001 seconds. This is far too short to observe EDL charging. The changing current with time seen in the experiment is due to the increasing/decreasing voltage of AC.*

Authors: We thank the reviewer for their comment. The references the reviewer points to involve a capacitive deionization (CDI) system, which is significantly different than ours. Specifically, CDI relies on highly porous, large surface area electrodes that enable high counter-ion adsorption, with the goal of desalinating a stream. When CDI electrodes are polarized, it can take a quite a while (hours) for ions to fully saturate the electrode's surface area (for multiple reasons, including the fact that diffusion in the confined spaces of the electrode is slow). In contrast, our system uses non-porous plate electrodes that attract a small number of ions (i.e., low capacitance), which leads to a short charging time. CDI electrodes must have a high capacitance to achieve desalination. In addition, our time-dependent current measurements (Fig. 4a,b and Fig. S6) unequivocally demonstrate that EDL charging in our system is achieved within our AC time-step.

To clarify these points, the following language was added to the manuscript:

(Pages 9-10, Lines 189-196)

“Note that our observed charging time scale is significantly lower than that observed in capacitive deionization (CDI) systems, which is proportional to electrode capacitance⁴⁴. Typical CDI electrodes have high specific capacitances (i.e., the ability to store charges), a

property that is critical to achieve desalination⁴⁵. Under these conditions, the time to fully charge the EDL is longer (i.e., greater EDL charging time scale). To support this statement, we compared the specific capacitance values of CDI systems to the specific capacitance of the electrodes used in our study and found the specific capacitance of CDI electrodes was generally orders-of-magnitude higher compared to our electrode (See Section S2 of the SI for details).”

(Pages S3-S4, Lines 67-76)

“Section S2. Specific capacitance in capacitive deionization

The specific capacitance (C_m , F g⁻¹) is determined by:

$$C_m = \frac{C_d}{m} \quad (S1)$$

where C_d (F) is the EDL capacitance and m (g) is the mass of the electrode. The electrode materials and specific electrode capacitance in the present study and CDI studies are shown in Table S2. The specific capacitance values of the CDI electrode (different forms of porous carbon) are consistently 3 orders-of-magnitude higher than that of our titanium electrode, which leads to the charging time in the scale of hours (while our charging time scale is fractions of a second).

Table S2. The comparison between the electrode in this study and the CDI electrodes.

Reference	Electrode materials	Specific capacitance (F g ⁻¹)	Charging time (s)
This study	Solid titanium plate	0.19	0.12
Dong et al. ⁴	Reduced Graphene Oxide/ Activated Carbon Nanofibers	256	6000
Laxman et al. ⁵	Activated carbon cloth/ ZnO nanorod	100	1000
Li et al. ⁶	Ordered mesoporous carbons	192	7200

”

Reviewer: 2. Lines 113-115, the statement ‘The incomplete EDL screens fewer surface charges, and leads to the creation of an alternating electric field that propagates far into the bulk and induces fast ion motion in the bulk fluid between the electrodes’ is confusing. AC leads to frequent change of electric field between the two electrodes, and thus includes fast ion motion. This effect could be direct. It does not have to be linked to EDL and surface charge (the authors claimed so since they used a hypothesis of a capacitor).

Authors: We appreciate the reviewer’s comments. While AC does induce fast ion motion due to the frequent changes in electric field, we believe that EDL charging is still relevant in our system. In our experiments, the ion motion dramatically decreased once the EDL was fully charged, so the EDL charging time (0.12 s) should be greater than AC time step to enable fast ion motion in the bulk. Furthermore, the surface charge of the electrode determines the strength of the electric field, which is reduced by the EDL formation. In addition, the observed decrease

in current during each AC half-period (Fig. 2a) represents the reduced ion motion and reduced electric field strength, consistent with the mechanisms of EDL charging and surface charge, respectively.

Critically, we observed the current and voltage exponentially decreased and increased, respectively, during EDL charging, which aligned with those of a typical RC (resistor-capacitor-in-series) circuit (Figs. 2a and S1a,b). Also, the measured current values were in good agreement with the predictions of our equivalent RC circuit model (Fig. S6). All these findings support our assumption of the electrode functioning as a capacitor.

Reviewer: 3. Conclusion, it was stated that ‘we introduce an electrically-driven method to significantly reduce the nucleation and growth of CaCO₃ crystals in super-saturated solutions’, and ‘Through the strategic application of AC potential, we offer a scalable and effective solution for extending the lifespan and reduced maintenance requirements for metallic surfaces in contact with super-saturated aqueous solutions (such as heat exchangers), thereby addressing a longstanding challenge in process engineering’. This method has already been introduced in a number of other works. Hence, such conclusions are misleading as the authors overstate their contributions.

Authors: We appreciate the reviewer’s feedback on the novelty of our method. While the application of AC conditions for anti-scaling purposes has been previously reported, our work distinguishes itself by addressing both homogeneous and heterogeneous nucleation of CaCO₃ crystals, as well as by proposing a mechanistic framework by which to understand the observed physical phenomena. To address the reviewer’s comment, we have included a detailed comparison of our study with previous AC scale mitigation research (Table S1) and revised the Introduction to reflect the primary difference.

Excerpts from the revised SI and manuscript that address this comment are reproduced below.

(SI, Pages S3, Lines 57-66)

“Section S1. Scope comparison of this study and previous AC scale prevention studies

The scope of this study is compared to previously published AC scale prevention studies. The relevant aspects of previous studies are summarized in Table S1. Critically, we explore the impact of AC potential on both homogeneous nucleation and heterogeneous nucleation, while other studies primarily are focused on the heterogeneous nucleation on the surfaces. By demonstrating its impact on homogeneous nucleation, we enable the possibilities to disrupt any crystal nucleation processes through application of alternating electric fields.

Table S1. The comparison between this study and other studies on the scope of investigation.

Reference	Nucleation	Electrical conditions	Mechanistic model	Application scenario
This study	Homogeneous nucleation, heterogeneous nucleation	Voltage, frequency	Quantitative model on equivalent circuit, ion	Heat exchangers

			displacement, and nucleation	
Rao et al. ¹	Heterogeneous nucleation	Frequency	Qualitative model	Membranes
Jung et al. ²	Heterogeneous nucleation	Single condition	Qualitative model	Membranes
Kim et al. ³	Heterogeneous nucleation	Voltage, frequency, waveform	Quantitative model on ion displacement	Membranes

(Page 4, Lines 89-93)

“The application of an alternating current (AC) to electrically conducting surfaces has been demonstrated to prevent surface crystallization during membrane desalination using electroactive membranes²⁴⁻²⁶. However, the mechanism behind the observed anti-scaling phenomena is not yet fully elucidated, and the observed scaling was assumed to be a result of heterogeneous precipitation only (see Section S1 of SI for details).”

(Page 5, Lines 113-118)

“At higher polarity reversal frequencies, the EDL transitions to a dynamic state, where it is constantly being formed and disrupted. The incomplete EDL screens fewer surface charges, and leads to the creation of an alternating electric field that propagates far into the bulk and induces fast ion motion in the bulk fluid between the electrodes³⁵, which can potentially impact both homogeneous and heterogeneous precipitation. It is this dynamic ion motion that can be responsible for the observed dramatic decrease in both surface and bulk crystal formation.”

Reviewer #2:

Reviewer: The manuscript presents an effective method to prevent CaCO₃ crystal nucleation and growth in supersaturated solutions using alternating electric fields (AC). The results and theory are convincing, demonstrating that AC disrupts ion collisions through rapid ion displacement induced by the charging and discharging of the electrical double layer (EDL) at certain frequencies. The study combines robust electrochemical characterization and modeling to demonstrate that high-frequency AC fields prevent both homogeneous and heterogeneous nucleation without significant electrode corrosion, suggesting practical applications in industrial systems like heat exchangers. This is a significant advancement in the field of mineral scale prevention. Below are some comments for the authors to consider.

Authors: We appreciate the reviewer’s positive assessment of our study and the recognition of its significance in mineral scale prevention. We carefully considered the reviewer’s comments and made revisions which we think significantly enhanced the quality of our manuscript.

Reviewer: Please discuss the novelty of this study further. The method of applying AC potentials to prevent mineral scaling, along with the underlying theory, has been presented in previous papers, including the authors' publication [1, 2].

Reference

[1] U. Rao, A. Iddya, B. Jung, C.M. Khor, Z. Hendren, C. Turchi, T. Cath, E.M. Hoek, G.Z. Ramon, D. Jassby, *Mineral scale prevention on electrically conducting membrane distillation membranes using induced electrophoretic mixing*, *Environ Sci Technol* 54(6) (2020) 3678-3690.
 [2] B. Jung, S. Ma, C.M. Khor, N.K. Khanzada, A. Anvari, X. Wang, S. Im, J. Wu, U. Rao, A.K. An, *Impact of polarity reversal on inorganic scaling on carbon nanotube-based electrically-conducting nanofiltration membranes*, *Chem Eng J* 452 (2023) 139216.

Authors: We appreciate the reviewer's comment. The primary novelty of this current study is the demonstration that the application of AC conditions can prevent homogeneous precipitation, as well as the development of a mechanistic framework that captures the physics responsible for the observed phenomena. Our study extends beyond the scope of previous research, which primarily addressed heterogeneous precipitation, and was empirically driven. By demonstrating that AC potentials can effectively prevent both homogeneous and heterogeneous precipitation, our work tackles a more fundamental aspect of nucleation and crystallization. This broader application of AC fields to disrupt crystal formation at an earlier stage represents a significant advancement in mineral scale prevention.

Excerpts from the revised manuscript and SI that address this comment are reproduced below.

(SI, Page S3, Lines 57-66)

“Section S1. Scope comparison of this study and previous AC scale prevention studies

The scope of this study is compared to previously published AC scale prevention studies. The relevant aspects of previous studies are summarized in Table S1. Critically, we explore the impact of AC potential on both homogeneous nucleation and heterogeneous nucleation, while other studies primarily are focused on the heterogeneous nucleation on the surfaces. By demonstrating its impact on homogeneous nucleation, we enable the possibilities to disrupt any crystal nucleation processes through application of alternating electric fields.

Table S1. The comparison between this study and other studies on the scope of investigation.

Reference	Nucleation	Electrical conditions	Mechanistic model	Application scenario
This study	Homogeneous nucleation, heterogeneous nucleation	Voltage, frequency	Quantitative model on equivalent circuit, ion displacement, and nucleation	Heat exchangers
Rao et al. ¹	Heterogeneous nucleation	Frequency	Qualitative model	Membranes

Jung et al. ²	Heterogeneous nucleation	Single condition	Qualitative model	Membranes
Kim et al. ³	Heterogeneous nucleation	Voltage, frequency, waveform	Quantitative model on ion displacement	Membranes

”

(Page 4, Lines 89-93)

“The application of an alternating current (AC) to electrically conducting surfaces has been demonstrated to prevent surface crystallization during membrane desalination using electroactive membranes²⁴⁻²⁶. However, the mechanism behind the observed anti-scaling phenomena is not yet fully elucidated, and the observed scaling was assumed to be a result of heterogeneous precipitation only (see Section S1 of SI for details).”

(Page 5, Lines 113-118)

“At higher polarity reversal frequencies, the EDL transitions to a dynamic state, where it is constantly being formed and disrupted. The incomplete EDL screens fewer surface charges, and leads to the creation of an alternating electric field that propagates far into the bulk and induces fast ion motion in the bulk fluid between the electrodes³⁵, which can potentially impact both homogeneous and heterogeneous precipitation. It is this dynamic ion motion that can be responsible for the observed dramatic decrease in both surface and bulk crystal formation.”

Reviewer: The turbidity, pH, and conductivity were monitored to indicate bulk precipitation of CaCO₃ within 2 hours. It may be helpful to elucidate the contribution of the dissolution of atmospheric CO₂ into the alkaline solution to the observed changes in pH and conductivity over 2 hours. Could the authors extend the duration of the experiment, which might allow for a clearer observation of the effects of AC potentials on CaCO₃ crystallization, as well as any potential long-term impacts of CO₂ diffusion? This could help differentiate between short-term and long-term effects and provide more comprehensive insights into the mechanisms.

Authors: We thank the reviewer for their suggestions. In response, we extended the experiment duration to 24 hours, and measured the turbidity, pH, and conductivity of CaCO₃ supersaturated solutions. The results demonstrated that the AC potential effectively mitigated the CaCO₃ crystallization over a prolonged period, and CO₂ diffusion into the solution did not affect the effectiveness of AC in preventing precipitation.

Excerpts from the revised SI and manuscript that address this comment are reproduced below.

(SI, Page S6-S7, Lines 110-136)

“Section S5. Long-term evaluation of CaCO₃ homogeneous precipitation prevention

We explored the long-term impact of AC potentials (i.e., 4 V_{pp}, 1 Hz) on CaCO₃ bulk precipitation. The bulk turbidity, pH, and conductivity of CaCO₃ supersaturated solution were measured over a 24-hour duration. The CaCO₃ bulk precipitation was successfully mitigated

over the long-term. Specifically, turbidity at 4 V_{pp}, 1 Hz was consistently orders-of-magnitude lower than the control, increasing from 0.17±0.01 to 0.29±0.02 NTU over the 2-to-24-hour period, compared to the control (0 V), where turbidity increased from 8.54±1.12 to 80.70±2.32 NTU (Fig. S3a). Critically, the bulk turbidity at 4 V_{pp}, 1 Hz reached only 0.29±0.02 NTU after 24 hours, indicating excellent performance even with a high supersatation index of 11.04.

During the 24-hour period, the conductivity (left y-axis) decreased due to both CO₂ diffusion and CaCO₃ precipitation, while the decline in pH (right y-axis) was primarily influenced by the CO₂ diffusion (Fig. 3b). Conductivity values during the first four hours of the experiment, decreased from 2518±13 to 2010±56 μS cm⁻¹ at 0 V, and from 2508±22 to 2087±43 μS cm⁻¹ at 4 V_{pp}, 1 Hz – statistically identical values. This suggested that the decline in conductivity was mainly due to CO₂ diffusion into the water. From 4 to 24 hours, conductivity values at 4 V_{pp}, 1 Hz decreased from 2060±40 to 1497±28 μS cm⁻¹ while at 0 V, they dropped from 1951±64 to 1313±58 μS cm⁻¹) (Fig. 3b). The more significant decrease in conductivity after 4 hours was caused by the CaCO₃ precipitation, as evidenced by the dramatic increase in bulk turbidity. In contrast, the pH values were similar between 4 V_{pp}, 1 Hz (from 11.97±0.02 to 10.76±0.06) and 0 V (from 11.95±0.02 to 10.64±0.09) over the 24-hour period, declining in a near-linear fashion (Fig. 3b). The decrease in pH reduced the concentration of CO₃⁻² for CaCO₃ precipitation, mitigating the initial sharp increase in bulk turbidity after 4 hours and leading to a plateau between 4 to 24 hours (Fig. 3a).

Fig. S3. a,b, Turbidity (a), pH and conductivity (b) as a function of time from 0 to 24 hours at 0 V and 4 V_{pp}, 1 Hz AC potential.”

(Page 14, Lines 286-291)

“We further extended the duration to 24 hours and evaluated the long-term impact of AC potential (i.e., 4 V_{pp}, 1 Hz) on turbidity, pH, and conductivity. The mitigation of homogeneous precipitation remained consistently effective and was not affected by CO₂ diffusion into the supersaturated solution (See Section S5 for details). Over the 24-hour period, the turbidity with AC potential increased only slightly from 0.09±0.01 to 0.29±0.03 NTU, while the control (0 V) saw a dramatic increase from 0.09±0.01 to 80.70±2.32 NTU (Fig. S3).”

Reviewer: On page 14, Figure 3d, the authors estimated the percentage of CaCO₃ electrode coverage and claimed that increasing the voltage from 4 to 20 V_{pp} at 100 Hz reduced CaCO₃ coverage (from 69±8% to 42±26%). However, the scales under 20 V_{pp} look denser than those under other experimental conditions. More quantification of the scales should be considered.

Authors: We thank the reviewer for their comment. To further characterize the CaCO₃ scale formed under the different conditions, we used ImageJ image analysis software to quantify the percentage (%) of the area occupied by CaCO₃ in SEM micrographs under different conditions. analysis shows very similar % area coverage and morphology of CaCO₃ crystals under the 4 V_{pp}, 100 Hz and 20 V_{pp}, 100 Hz conditions. Furthermore, we used contact angle measurements to further characterize scaling on the electrode surfaces. We modified the text to include this data:

(SI, Pages S7-S9, Lines 138-164)

“Section S6. Additional surface characterization data

To investigate the CaCO₃ coverage at a microscopic level, we compared SEM micrographs under the conditions of 4 V_{pp}, 100 Hz and 20 V_{pp}, 100 Hz (where the scale appears densest) as an example. We used image processing software (ImageJ) to quantify the percentage area coverage (%) of CaCO₃, calculated by:

$$\% \text{ Area} = \frac{S_{\text{crystal}}}{S_{\text{image}}} \quad (\text{S2})$$

where S_{crystal} is the area of CaCO₃ crystals and S_{image} is the area of the SEM micrograph. The average CaCO₃ % area was nearly identical under these conditions (7.06±0.76% for the 4 V_{pp}, 100 Hz and 6.93±1.31% for 20 V_{pp}, 100 Hz (Fig. S4a). Also, the respective SEM micrographs showed very similar crystal structure and distribution (Fig. S4b).

In addition, surface contact angle measurements were used to quantify CaCO₃ scaling under different electrical conditions (Fig. S4c). The contact angle of the bare titanium surface (c1 in Fig. S4c) was measured to be 87.3±0.9°, which is consistent with previously reported values⁷. Calcite, with a lower contact angle (48° to 60°)⁸, leads to a reduction in the contact angle due to CaCO₃ buildup, when present. The measured contact angles (c2-c9 in Fig. S4c) correspond well with the observed CaCO₃ coverage (Fig. 3d). Specifically, a higher contact angle of 75.2±2.7° was observed at 20 V_{pp}, 100 Hz, compared to 71.4±1.3° at 4 V_{pp}, 100 Hz, suggesting reduced CaCO₃ coverage at higher voltages. Also, nearly identical contact angles of 82.1±4.1°, 80.7±3.5°, 83.6±2.0° at 4 V_{pp} for 0.1, 1, and 10 Hz, respectively, suggest minimal CaCO₃

coverage at these conditions. The low contact angles of $67.1 \pm 1.8^\circ$ and $71.4 \pm 1.3^\circ$ at 2 V and 4 V_{pp}, 100 Hz, respectively, are comparable to the $65.2 \pm 2.2^\circ$ at 0 V, indicating a high CaCO₃ coverage similar to that seen in the absence of an applied potential.

Fig. S4. a,b,c, Area % of CaCO₃ under 4 V_{pp}, 100 Hz and 20 V_{pp}, 100 Hz conditions (a), and SEM micrograph at 4 V_{pp}, 100 Hz and 20 V_{pp}, 100 Hz, at the same magnification (5000x) as that of Fig. 3e in the manuscript (b). The contact angles of the electrodes (c1-c9) under different electrical conditions (c).”

(Page 15, Lines 307-317)

“Scanning electron microscope (SEM) images and energy-dispersive X-ray spectroscopy (EDS) mapping were used to compare the structure and elemental composition of electrode surfaces under open-circuit (0 V) and 4 V_{pp}, 1 Hz conditions (Fig. 3e). SEM images show that at 0 V, the surface was covered by cubic-like calcite crystals. In contrast, when 4 V_{pp}, 1 Hz potentials were applied, very few CaCO₃ crystals were visible, and only titanium was detected. In addition, the % area of CaCO₃ (defined as the ratio of the area of CaCO₃ crystals to the area of the SEM micrograph) was found to be consistent when comparing the SEM micrographs of

the same magnification across different electrical conditions (Fig. S4a,b). The pure titanium signal in the overall EDS mapping image further confirms the inhibition of CaCO₃ crystal formation (Fig. 3e). The individual EDS mapping results and spectrum showed a clear distribution of all the elements on the electrode surface (see Section S6 for details).”